# An INS-UWB Based Collision Avoidance System for AGV

**Shunkai Sun [1,2], Jianping Hu [1,*], Jie Li [2], Ruidong Liu [2], Meng Shu [2] and Yang Yang [3]**

[1] Faculty of Electrical Engineering and Computer Science, Ningbo University, Ningbo 361102, China; shunkaisun@126.com

[2] Ningbo Cigarette Factory, China Tobacco Zhejiang Industrial Co., Ltd., Hangzhou 315500, China; lijie@zjtobacco.com (J.L.); liuruidong@zjtobacco.com (R.L.); shumeng@zjtobacco.com (M.S.)

[3] International Business Department, China Tobacco Zhejiang Industrial Co., Ltd., Hangzhou 310008, China; yy@zjtobacco.com

* Correspondence: hujianping2@nbu.edu.cn

**Abstract:** As a highly automated carrying vehicle, an automated guided vehicle (AGV) has been widely applied in various industrial areas. The collision avoidance of AGV is always a problem in factories. Current solutions such as inertial and laser guiding have low flexibility and high environmental requirements. An INS (inertial navigation system)-UWB (ultra-wide band) based AGV collision avoidance system is introduced to improve the safety and flexibility of AGV in factories. An electronic map of the factory is established and the UWB anchor nodes are deployed in order to realize an accurate positioning. The extended Kalman filter (EKF) scheme that combines UWB with INS data is used to improve the localization accuracy. The current location of AGV and its motion state data are used to predict its next position, decrease the effect of control delay of AGV and avoid collisions among AGVs. Finally, experiments are given to show that the EKF scheme can get accurate position estimation and the collisions among AGVs can be detected and avoided in time.

**Keywords:** AGV; INS; UWB; collision avoidance

---

## 1. Introduction

Due to the development of industrial technology, AGV has been widely used in many applications [1,2]. Positioning technology is an important technology of AGV. However, traditional positioning technologies have some drawbacks, such as hardware requirements, poor flexibility and difficulties with complex working environments [3].

Despite many navigation methods for AGV having been proposed, accurate positioning is still a very important task [4]. The electromagnetic-based guidance system for AGV [5] presents some inconvenience such as maintenance and modifications. As an effective autonomous navigation technology, INS provides complete navigation information but requires external information to provide high positioning accuracy [6]. Laser-based navigation systems provide not only the position of the AGV but detect the presence of obstacles in the path [7]. However, they cannot be used in non-line-of-sight (NLOS) scenarios. Vision-based navigation systems obtain information of the AGV surroundings by means of image processing techniques [8], providing high positioning accuracy of a larger sensing area. However, the expensive computational cost demanded in the image analysis make them ill-suited for real-time applications.

On the other hand, AGV positioning and obstacle detection have also been addressed by wireless networks. According to different RF protocols, indoor positioning based on WSN can be classified into Radio Frequency Identification (RFID) [9], *WIFI* [10], *ZigBee* [11], *Bluetooth* [12] and *UWB* [13].

UWB technology is based on sending and receiving carrier-less radio impulses using extremely accurate timing and it is particularly suitable for distance estimation and positioning applications. However, the noise reduction cannot be conducted effectively, and the results are always noisy [14]. Therefore, an estimator is used to estimate state variables from noisy measurement [15] and numerous filtering. Thus, numerous filtering methods are applied with various localization techniques [16,17]. The most popular techniques are based on Kalman filters, especially on the Extended Kalman Filter [18].

In order to avoid collisions, an important issue in the operation and control of AGV is the obstacle detection where a possible solution could be the use of predetermining routes of the AGV. In this paper, we present a collision avoidance system for AGV. The method integrates both INS and UWB to get the position of both the AGV and the obstacle as well as the pose of the former. Compared with previous works, the proposed system improves the system flexibility, reducing costs and complexity, as well as expanding the use in different scenarios. The paper is organized as follows:

In Section 2, the system scenario is described and the environment that AGV works in is presented. The system overview is presented in Section 3 that details the research points in this paper. The experiments and results of the research are shown in Section 4. The conclusions are given in Section 5.

## 2. Factory Scenario of the Application

The factory environment where the AGV works is shown in Figure 1. The AGV requires information of its location and the position of the shelves and possible obstacles in order to avoid collisions.

In the factory warehouse environment, the navigation and collision avoidance are carried out with an electromagnetic guidance system which requires fixed paths; thus, the flexibility of the system is poor. AGV with UWB technology could achieve a high absolute positioning result with high flexibility. UWB integrated with INS and other sensors could be applied to the collision avoidance and path planning of AGV in the factory warehouse environment.

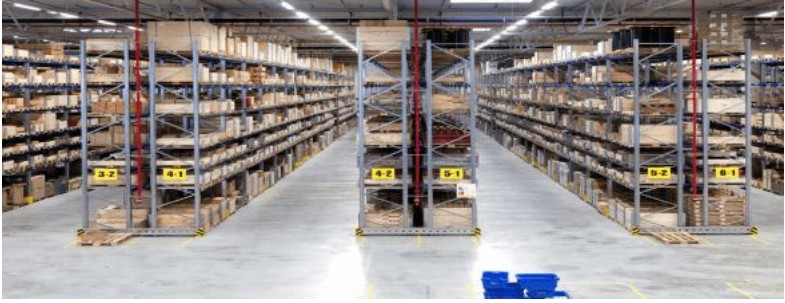

**Figure 1.** Warehouse of the factory.

## 3. Collision Avoidance System Design

### 3.1. UWB Positioning

The UWB localization of the AGV collision avoidance system is based on three positioning points. Considering the traditional factory warehouse environment, the UWB anchor nodes are deployed on the corners of the warehouse shelves to decrease NLOS (none line-of-sight) effects as shown in Figure 2. After deploying the UWB anchor nodes in the warehouse, an electronic map with a coordinate will be established, which will accurately note each UWB anchor nodes and obstacles. With the deployed UWB nodes and the electronic map, the AGV could get its coordinate and calculate the distances between the shelves and obstacles in the path for collision avoidance.

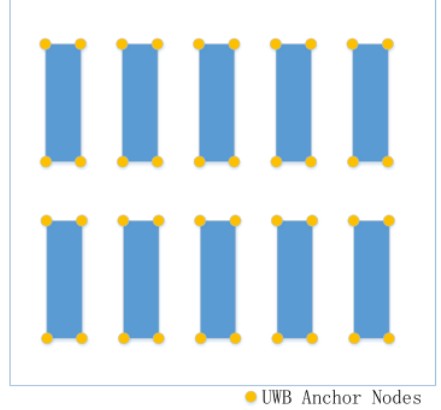

**Figure 2.** Deployment of UWB anchor nodes.

The movement of AGV is in a 2D plane. whereas the UWB nodes are commonly installed in high places in order to decrease the interferences of NLOS. The solution is the projection of the anchor nodes onto the ground plane, as shown in Figure 3. The smaller the distance, the higher the accuracy of UWB ranging; thus, the three anchor nodes closest to the AGV are utilized in its positioning.

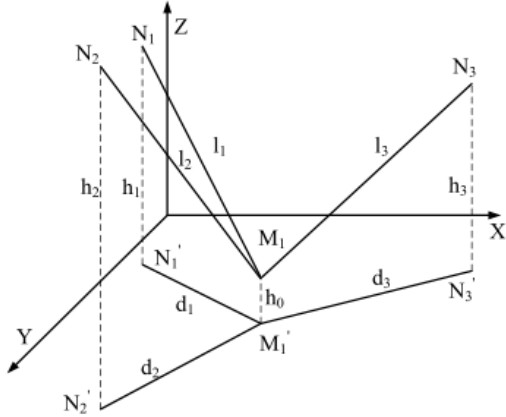

**Figure 3.** Range projection.

Denoting the three UWB anchor nodes as *N1*, *N2*, *N3*, the distances between the AGV and the three anchor nodes as $(l_1, l_2, l_3)$, the heights of the anchor nodes as $(h_1, h_2, h_3)$, and the height of the UWB device on the AGV as $h_0$, the vertical projections of the three UWB anchor nodes $(d_1, d_2, d_3)$ are

$$
\begin{bmatrix} d_1 \\ d_2 \\ d_3 \end{bmatrix} = \begin{bmatrix} \sqrt{l_1^2 - (h_1 - h_0)^2} \\ \sqrt{l_2^2 - (h_2 - h_0)^2} \\ \sqrt{l_3^2 - (h_3 - h_0)^2} \end{bmatrix} . \tag{1}
$$

The locations in the coordinate system of the anchor nodes *N1*, *N2*, *N3* and the AGV node *D* are $(x_1, y_1, (x_2, y_2), (x_3, y_3), (x, y)$, respectively. Thus,

$$
\left\{ \begin{array}{l} (x - x_1)^2 + (y - y_1)^2 = d_1^2 \\ (x - x_2)^2 + (y - y_2)^2 = d_2^2 \\ (x - x_3)^2 + (y - y_3)^2 = d_3^2 \end{array} \right\} . \tag{2}
$$

If $\Delta = (x_1 - x_2)(y_1 - y_3) - (x_1 - x_3)(y_1 - y2) \neq 0$, the linearization method could be adopted to get a coordinate of the point $D$:

$$\begin{bmatrix} x \\ y \end{bmatrix} = A^{-1}b = \begin{bmatrix} \frac{K_1}{K}(y_2 - Y_3) + \frac{K_2}{K}(y_3 - y_1) \\ \frac{K_1}{K}(x_3 - x_2) + \frac{K_2}{K}(x_1 - x_3) \end{bmatrix}, \tag{3}$$

$$K = 2(x_1 - x_3)(y_2 - Y_3) - 2(x_2 - x_3)(y_1 - y_3), \tag{4}$$

$$K_1 = x_1^2 - x_3^2 + y_1^2 - y_3^2 - d_1^2 + d_3^2, \tag{5}$$

$$K_2 = x_2^2 - x_3^2 + y_2^2 - y_3^2 - d_2^2 + d_3^2. \tag{6}$$

The coordinate of unknown point $D(x, y)$ can be calculated. The three point positioning method is an easy method;however, due to the distance estimation error in real applications, it may become an unsolved problem. There are several conditions which could result in unsolved problems such as those illustrated in Figure 4a,b which are the most common. In order to get better localization results, the UWB ranging errors should be addressed.

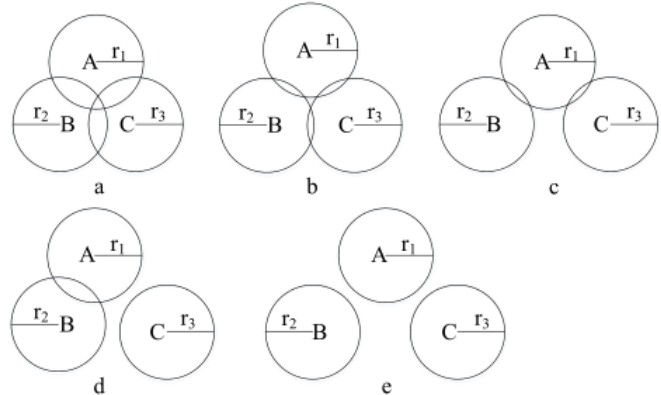

**Figure 4.** Possible error conditions.

### 3.2. INS and UWB Integrated Positioning Method

In the UWB distance estimation, the obstacle between the UWB transmitter and receiver has to avoid the line-of-sight propagation of the UWB signal. The receiver could only get the signal through diffraction, refraction and reflection, which will generate NLOS range estimation errors. The estimated distance in NLOS conditions is inaccurate, which makes the distance a biased estimate. The UWB distance estimating data shall be processed before being used in the positioning method. EKF (extend Kalman filter) which integrates INS and UWB is adopted in this paper to deal with the errors in the range estimation and positioning of the AGV.

The Kalman filter adopted in this paper processes the UWB ranging distance data. Both the state and observation equations are as follows:

$$X(k+1) = \Theta X(k) + \Gamma u(k) \tag{7}$$

$$Z(k) = HX(k) + v(k) \tag{8}$$

$$\Theta = \begin{bmatrix} 1 & T & 0 & 0 \\ 0 & 1 & 0 & 0 \\ 0 & 0 & 1 & T \\ 0 & 0 & 0 & 1 \end{bmatrix}; \Gamma = \begin{bmatrix} \frac{T^2}{2} & 0 \\ T & 0 \\ 0 & \frac{T^2}{2} \\ 0 & T \end{bmatrix}; H = \begin{bmatrix} \frac{1}{2} & 0 \\ 0 & 0 \\ 0 & 0 \\ 0 & 1 \end{bmatrix}; X = \begin{bmatrix} x \\ x \\ y \\ y \end{bmatrix}; Z = \begin{bmatrix} x \\ y \end{bmatrix}, \tag{9}$$

where $u, v$ are the observation and process noise, respectively, and $T$ is the sampling period.

The mean square error of the next step prediction is:

$$\widehat{X}_{k|k-1} = \Theta_{k,k-1}\widehat{X}_{k-1} \tag{10}$$

and the state estimation is:

$$\widehat{X}_k = \widehat{X}_{k|k-1} + K_k(Z_k - H_k\widehat{X}_{k|k-1}), \tag{11}$$

$$K_k = P_{k|k-1}H_k^T(H_k P_{k|k-1}H_k^T + R_k)^{-1}. \tag{12}$$

The mean square error of next step prediction is:

$$P_{k|k-1} = \Theta_{k|k-1}P_{k-1}\Theta_{k|k-1}^T + \Gamma_{k-1}Q_k\Gamma_k^T \tag{13}$$

and the estimated mean square error is:

$$P_k = (I - K_k H_k)P_{k|k-1}. \tag{14}$$

In the proof of Kalman filtering, given the initial system state error $\widehat{X}_0$ and initial state error variance matrix $\widehat{P}_0$, the predicted ranging data $\widehat{X}_k$ ($k = 1, 2, 3, ...$) of time $k$.

The gyroscope is one of the key devices in INS localization and the main error is due to the gyroscope drifting. In a short time, the white and quantizing noises cause an important effect on the angular error. The modeling of the gyroscope mainly includes: constant and random errors and the model of the gyroscope is:

$$\varepsilon = \varepsilon_e + \varepsilon_r + \omega_g. \tag{15}$$

In the design of the integrated localization method, the biased error $\Delta_i$ ($i = x, y, z$) is considered in the error model of the accelerometer:

$$\Delta_i = \Delta_{bi} + \omega_{ai} \quad (i = x, y, z), \tag{16}$$

where $\Delta_{bi}$ is the constant error and $\omega_{ai}$ is the white noise.

The error of the INS localization includes the error of the device, which consist of the scale and the installation errors.

The principle of the EKF is to linearize the system, transfer the system model to state equation, measure the equation error and predict the state error with the standard Kalman filter.

Assume that the nonlinear observation model is:

$$Z(t) = h(X(t)) + V(t), \tag{17}$$

where $h(x)$ is the nonlinear function of the state vector.

Then, the discrete form is:

$$\delta Z_k = h(X_k) - h(\widehat{X}_k) = H_k \delta X_k + V_k, \tag{18}$$

$$H_k = \frac{\partial h}{\partial X}, \quad X = \widehat{X}_{k|k-1}. \tag{19}$$

After linearization, standard Kalman filtering is used to calculate both the error state and the observation models.

### 3.3. AGV Collision Avoidance System

Based on the accurate localization, the workflow of the UWB and INS based AGV collision avoidance system is shown in Figure 5. After the AGV UWB node sends a ranging request, the UWB anchor nodes on the warehouse shelves will receive the signal and reply to it. Through TDOA

(Time Difference of Arrival), the range between the AGV and the anchor nodes could be estimated. Then, the AGV might receive data from various anchor nodes, and could estimate the distances from them:

$$R = \begin{bmatrix} r_1, & r_2, & r_3, & ..., & r_n, \end{bmatrix} \tag{20}$$

where $r_n$ represents the current estimated distance between AGV and anchor node $n$.

In order to reduce the computational complexity, the nearest three UWB anchor nodes are considered as reference nodes in the positioning of the AGV. In addition, the estimated distance from the AGV and the three anchor nodes are:

$$R = \begin{bmatrix} r_{N1}, & r_{N2}, & r_{N3} \end{bmatrix}. \tag{21}$$

At the same time, the system will record the node id of the three UWB anchor nodes and find the corresponding location coordinates according to the pre-established electronic map. The coordinates could be represented as follows:

$$A_n = [x_{pn}, y_{pn}] \quad n \in \{1, 2, 3\}. \tag{22}$$

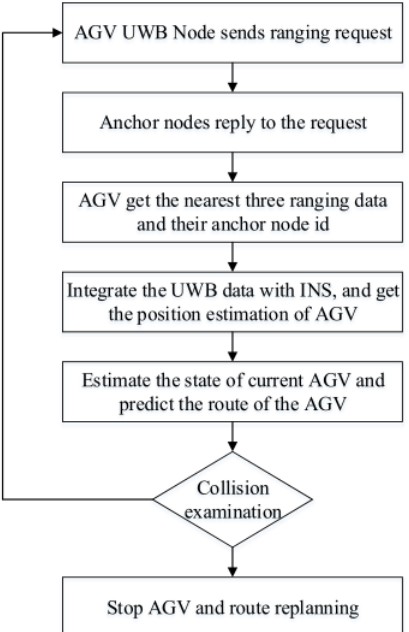

**Figure 5.** Flowchart of collision avoidance.

With the UWB ranging result $R_N$ and the coordinates of the UWB anchor nodes $P_n$, the accurate localization result could be achieved with EKF integrating the INS data. The coordinate of the AGV at time k can be represented with $[x_k, y_k]$.

In order to avoid the collision of the AGV, the real-time distance between the coordinate of the AGV and the anchor nodes should be measured. The movement of the AGV shall be predicted at the same time to prevent the collision resulting from the control delay of the AGV. The movement prediction of the AGV could be accomplished with the positioning estimation, direction of motion and current velocity. The direction and velocity information could be obtained from the INS sensors.

As shown in Figure 6, there could be several kinds of collision situations, no collision (a), about to collide (b), and will have collision (c).

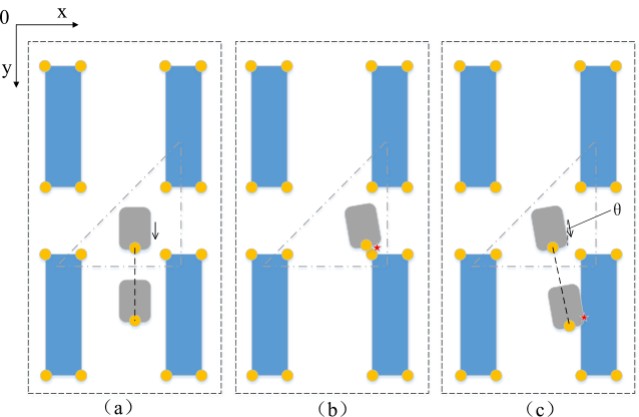

**Figure 6.** Different collision avoidance situations.

In the warehouse scenario, once the position coordinate $[x_k, y_k]$, the velocity $v_{pn}$ and the angel $\theta$ between the AGV and $+y$ direction of the coordinate are known, the left and right side position can be calculated with the size of AGV as follows:

$$\begin{bmatrix} l_{x_k} \\ l_{y_k} \end{bmatrix} = \begin{bmatrix} x + \frac{w}{2}\cos\theta \\ y - \frac{w}{2}\sin\theta \end{bmatrix} ; \begin{bmatrix} r_{x_k} \\ r_{y_k} \end{bmatrix} = \begin{bmatrix} x - \frac{w}{2}\cos\theta \\ y + \frac{w}{2}\sin\theta \end{bmatrix}, \tag{23}$$

where $w$ is the width of the AGV.

Considering the control delay of the AGV, assuming that the AGV keeps its current state for time $\Delta t$, the position of the left and right side of the AGV could be calculated and predicted.

If the area covering the four points $[l_{x_k}, l_{y_k}]$, $[r_{x_k}, r_{y_k}]$, $[l_{x_{k+1}}, l_{y_{k+1}}]$, $[r_{x_{k+1}}, r_{y_{k+1}}]$ does not contain obstacles in the electronic map, no collision happens. When the area has obstacles, collisions will happen and the AGV adjusts its direction until it meets the demand for non-collision.

## 4. Experiment Results

In this section, the performance of the proposed UWB and INS based AGV collision avoidance system is tested, including the UWB and INS positioning accuracy test and the collision avoidance ability test of the AGV.

The AGV adopted in the experiment is a Discovery *Q2* four omnidirectional wheel AGV, as shown in Figure 7. It is a small and multi-port AGV high reliability robot system.

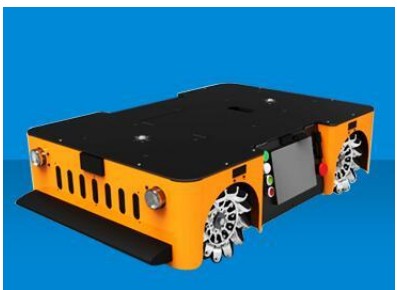

**Figure 7.** Outlook of AGV.

Firstly, the positioning accuracy of UWB is tested. Due to the limitations of the test site, the test ground is set as a $10 \times 10$ m area. As shown in Figure 8a, the circles represent the UWB anchor nodes, and the triangle represents the sample test locations. At the test locations, the AGV collects several sets of ranging and positioning data to assess the UWB positioning accuracy. The error distribution is shown in Figure 8b. It is obvious that the test points closer to the center of the three UWB anchor nodes achieve smaller positioning error. According to the experimental results, when the UWB anchor

nodes are far from each other, the positioning accuracy is poor. The results are meaningful for the deployment of the UWB anchor nodes and users can choose the distance of the anchor nodes according to their demanded accuracy.

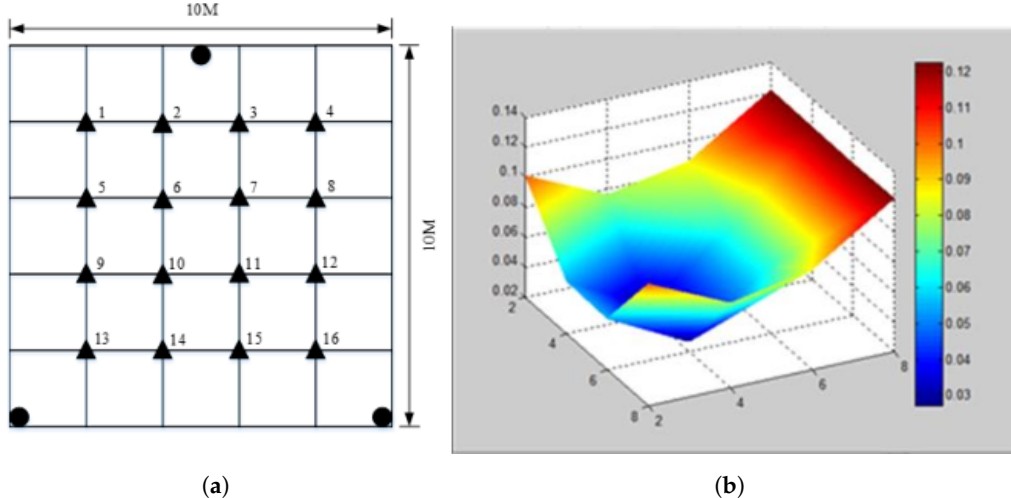

(**a**)　　　　　　　　　　　　　　　　　　　　　　　　(**b**)

**Figure 8.** (**a**) UWB localization accuracy test; (**b**) error distribution.

In the test, the positioning result on each test point is recorded 10 times and the average localization error is shown in Table 1. At the same time, the error distribution is shown in Figure 9.

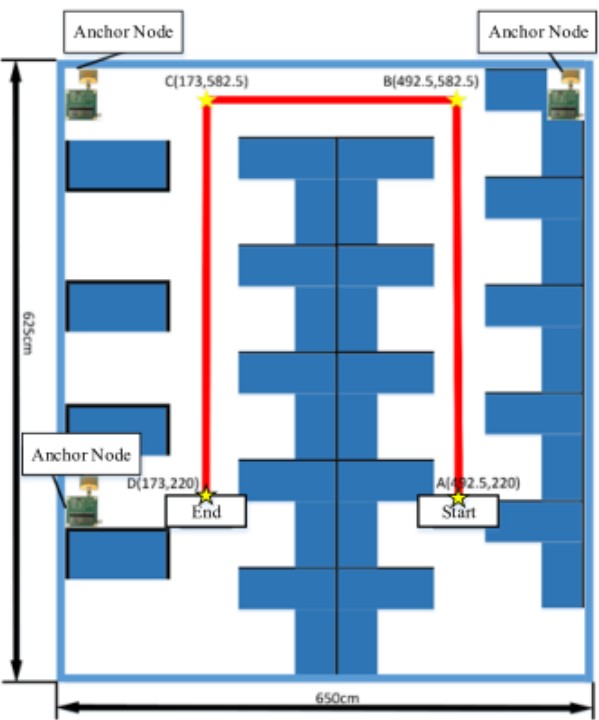

**Figure 9.** Map of warehouse environment.

**Table 1.** Location error in different positions

| Line 1 | 0.1226 | 0.1219 | 0.1222 | 0.1235 | 0.1226 |
|---|---|---|---|---|---|
| Line 2 | 0.0871 | 0.0664 | 0.0662 | 0.0861 | 0.0765 |
| Line 3 | 0.0773 | 0.0272 | 0.0291 | 0.0755 | 0.0528 |
| Line 4 | 0.1017 | 0.0546 | 0.0527 | 0.1001 | 0.0773 |
| Average | 0.0972 | 0.0675 | 0.0676 | 0.0963 | 0.0823 |

After UWB positioning accuracy test, the positioning of UWB and INS integration is also tested. A map of the warehouse environment is first established as shown in Figure 9. Due to the hardware limitations, three UWB anchors are used in the test environment. The AGV goes from Start point to the End point, and goes through route points A, B, C, and D to test the accuracy on the path.

Three positioning methods including UWB, INS and INS-UWB are tested and compared. The results are shown in Figure 10.

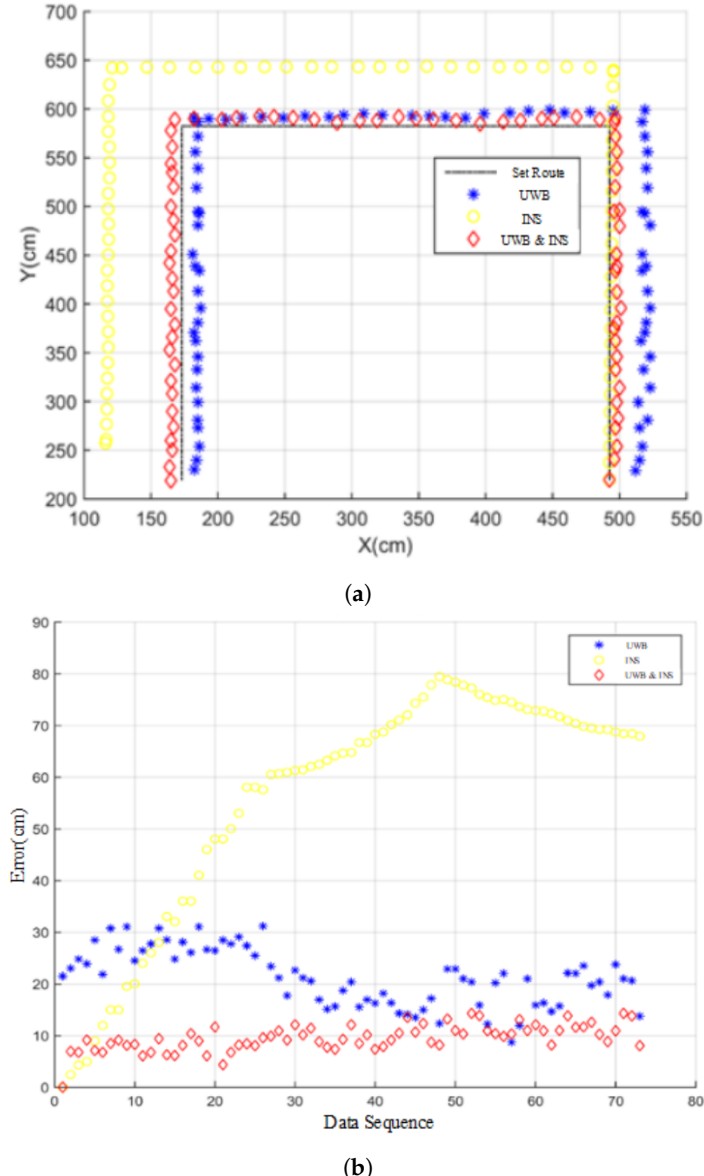

(a)

(b)

**Figure 10.** (**a**) route comparison method; (**b**) error comparison of different localization methods.

As shown in Figure 10, the positioning error of UWB is relatively stable in the whole route, which is between 10 cm and 30 cm. As the anchor nodes are closer to point B, C and D, the error in AB period is larger than that in BC and CD period, which corresponds to the result in UWB positioning error test. The error of the INS positioning is smaller in the first period and larger in the middle and last periods. The INS-UWB positioning method has the advantages of both positioning methods. The overall error is smaller than both single positioning methods and it is relatively stable, less than 15 cm, meeting the requirement for AGV positioning.

In the AGV collision avoidance test, the AGV is instructed to across the warehouse with shelves in order to reach a destination. The collision avoidance effect is shown in Figure 11 where the black blocks are the obstacles and the red line is the final route of the AGV. According to the results, the route in the blue circles represents that the AGV has detected the obstacles and it has correctly adjusted its direction to avoid the collision. The experimental results show that the INS-UWB based AGV collision avoidance system can be used in environments with fixed obstacles.

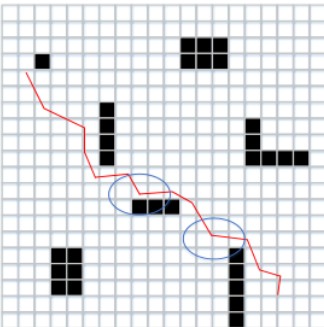

**Figure 11.** Collision avoidance effect.

## 5. Conclusions

Compared to traditional inertial- and laser-based guiding systems, the proposed method is more flexible and cheaper with good accuracy and stability. In order to realize an accurate positioning, an electronic map of the warehouse is established where the UWB anchor nodes are deployed. The location coordinate of the AGV is obtained by three nearby UWB anchor nodes in the map. An EKF method which integrates INS and UWB data is adopted to improve the positioning accuracy. In order to avoid collisions, the current location of the AGV and its motion state data are utilized to predict its next position to decrease the effect of control delay of the AGV. Experimental results show that the method proposed in this paper achieves accurate positioning estimation and the AGV effectively detects obstacles and avoids possible collisions, thus ensuring the safety of the AGV.

**Author Contributions:** Y.Y. contributed conceptualization of paper; J.H. contributed experiments data curation and experiments supervision; R.L. performed the project administration and visualization; J.L. analyzed the experiments data; S.S. designed the algorithm, performed the experiments; S.S. and M.S. wrote the paper.

**Funding:** This research received no external funding.

**Conflicts of Interest:** The authors declare no conflict of interest.

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
