# Peer review of "An INS-UWB Based Collision Avoidance System for AGV"

_algorithms, doi:10.3390/a12020040_

Round 1

Reviewer 1 Report

This paper presents a collision avoidance system for automated guided vehicles in factories. The proposed system combines inertial sensors and ultra wide band radio frequency. Inertial sensors are used to determine the position, speed, direction of the vehicle, whereas ultra wide band technology is used to detect the presence of an object and its distance to the vehicle. The main contribution is in the used of the extended Kalman filter which makes the system improve in terms of localization accuracy, cost and complexity.

The work presented in this paper is interesting, the overall merit is clear and the results are promising. However the presentation of the work is not good.

The document is written in a careless manner.

- page 2, line 62: “… UWBanchor…“

- page 3, lines 6-7: ” … he coordinate system are given, They are…”

- page 4, line 1: “… common conditions, In order to ,,,”

- page 6, line 15-16: “…and predicted: If the area ,,,”

- page 7, line 17: “…in Table ??.”.

By the way, there is no table in the paper.

The paper presents a large quantity of grammatical errors and the english style is not suitable for a journal as Algorithms. I completely understand the difficulty of writing in a foreign language thus, I encourage the authors to be assisted by an English-speaker.

I strongly encourage the authors to carefully rewrite the paper and resubmit the new version.

Author Response

Dear  Reviewers:

Thank you for the comments concerning our manuscript entitled An INS-UWB based collision avoidance system for AGV. Those comments are all valuable and very helpful for revising and improving our paper, as well as the important guiding significance to our researches. We have studied comments carefully and have made correction which we hope meet with approval. The main corrections in the paper and the responds to the reviewers’ comments are as following:

Recommend to correct line 62 in page 2, lines6-7 in page 3,  line 1 in page 4, lines15-16 in page 6, line 17 in page.

Response:
  It is our negligence and we are sorry about this. According to comment, related content have been improved. 

    (1) Correct something wrong lines in line62 of page 2, lines6-7 of page 3, line 1 of page 4, lines 15-16 of page 6, line 17 of page 7.  

     (2) Supply a table  in page 8.

                                                    2019-01-21

Reviewer 2 Report

The author presents in the paper system for AGV. It is good paper with trasparently resultates and good logical standard.

I recommend you insert any European authors in the Introduction, which deal in AGV. For example: http://www.mmscience.eu/content/file/archives/MM_Science_201690.pdf

I recommend you to explain the connection point M1 of the three UWB anchor nodes N1, N2, N3, in Figure 3.

I recommend you make clear and visible written variables, information (data) in Figures (now it is so small and can not read):
Figure 3.
Figure 4.
Figure 9.
Figure 10.

I recommend you make bigger Figure 5. with algorithm (now it is so small and can not read)

I recommend very carrefully read all text and correct typography in all the text, for example
70 (a), (b) not (a),(b)
70 conditions. not conditions,
80 EKF (extend not EKF(extend
121 Fig. 8 (a), or 124 Fig. 8(b)
142 15 cm not 15cm
etc

After minor revision recommend it to publish.

Author Response

Dear  Reviewers:

Thank you for your comments concerning our manuscript entitled An INS-UWB based collision avoidance system for AGV. Those comments are all valuable and very helpful for revising and improving our paper, as well as the important guiding significance to our researches. We have studied comments carefully and have made correction which we hope meet with approval. The main corrections in the paper and the responds to the reviewers’ comments are as following:

(1) Recommend to insert any European authors in the introduction, which deal in AGV.

(2) Recommend to explain the connection point M1of the three UWB anchor nodes N1,N2,N3 in figure 3.

(3) Recommend to make clear and visible written variables, information in figures.

(4) Recommend to make bigger figure 5.

(5) Recommend to correct typography in all the text. For example lines 70 ,80, 121, 142.

Response:
  It is our negligence and we are sorry about this. According to comment, related content have been improved.
  (1)insert the an European author in the introduction,the authors research paper link http://www.mmscience.eu/content/file/ archives/ MM_Science_ 201690.pdf.

(2) Point M1 is top of UWB. The connections N1, N2, N3 are distances between M1 and three UWB anchor nodes. This is explained in line 71.

(3) Correct all figures size to make clear and visible written variables.

(4) Make the figure 5 bigger.

(5) Correct typography in lines 70, 80, 121, 142.

                                                    2019-01-21

Round 2

Reviewer 1 Report

In the first revision of this article, it was recommended to revised the English grammar. The presentation of the paper was really awful thus, I did not addressed any comment about the proposed method. In this revised version, the authors have not done any revision.

This article should be rejected for two reasons: 1º the English used is not correct. The paper presents errors practically in every line. 2º The authors do not demonstrate any interest in their own work.

The recommendation for this revised version which is identical to the first one, is to be accepted. I recommend to include the next modifications:     

Abstract:

line 3: … such as inertial and laser guiding have low …

line 4: … environmental requirements.

line 5: … is introduced to improve …

line 6: … map of the factory …

line 7: … to realize an accurate positioning.

line 10: … experiments are given …

1. Introduction:

line 16: … drawbacks such as hardware requirements, poor flexibility and difficulties with complex working enviroments [3].

line 18: … Despite many navigation methods for AGV have been proposed, accurate positioning is still a very important task [4].

line 20: … The electromagnetic-based guidance system for AGV [5] presents some inconvenients such as maintenance and modifications.

line 22: … INS provides complete navigation information but requires external information to provide high positioning accuracy [6].

line 23: Laser-based guidance systems provide not only the position of the AGV but detect the presence of obstacles in the path [7]. However, they cannot be used in non-line-of-sight (NLOS) scenarios.

line 23: Vision-based navigation systems obtain information of the AGV surroundings by means of image processing techniques [8], providing high positioning accuracy of a larger sensing area. However, the expensive computational cost demanded in the image analysis make them ill-suited for real-time applications.

line 29: On the other hand, AGV positioning and obstacle detection have also been addressed by wireless networks.

line 31: … Bluetooth [12] and UWB [13].

line 34: … , a estimator is used to estimate state variables

line 36: … from noisy measurement [15] and numerous filtering …

line 36: … are based on Kalman filters, …

line 38: In order to avoid collisions, an important issue in the operation and control of AGV is the obstacle detection where a possible solution could be the use of predetermining routes of the AGV.

line 40: In this paper we present a collision avoidance system for AGV. The method integrates both INS and UWB to get the position of both the AGV and the obstacle as well as the pose of the former.

line 42: Compared with previous works, the proposed system improves the system flexibility, reducing costs and complexity, as well as expanding the use in different scenarios.

2. Scenario of factory application

line 48: Factory scenario of the application

line 49: The factory enviroment where the AGV works is shown in Fig. 1. The AGV requieres information of its location and the position of the shelves and possible obstacles in order to avoid collisions.

line 53: … are carried out with an electromagnetic guidance system which requieres fixed paths thus, the flexibility of the application is poor.

line 57: Figure 1. Warehouse of the factory

3. Collision avoidance system design

line 60: The UWB localization of the AGV collision avoidance system is based on three positioning points.

line 63: … in the warehouse, an electronic map …

line 66: … shelves and obstacles in the path for collision avoidance.

line 67: The movement of AGV is in a 2D plane whereas the UWB nodes are commonly installed in high places in order to decrease the interferences of NLOS.

line 68: The solution is the projection of the anchor nodes onto the ground plane, as shown in Fig. 3.

line 69: The smaller distance the higher accuracy of UWB ranging thus, the three anchor nodes nearest to the AGV are utilized in the its positioning.

line 71: Denoting the three UWB anchor nodes as N1, N2, N3, the distances between the AGV and the three anchor nodes as (l1, l2, l3), the heights of the anchor nodes as (h1, h2, h3), and the height of the UWB device on the AGV as h0, the vertical projections of the the three UWB anchor nodes (d1, d2, d3) are,

line ?: The locations in the coordinate system of the anchor nodes N1, N2, N3 and the AGV node D are (x1, y1), (x2, y2), (x3, y3), (x, y), respectively thus,

line ?: … coordinate of the point D:

line 72: The coordinate of the unknown point D(x, y) can be calculated. The three point positioning method is an easy method however, due to the distance estimation error in real applications, it may become an unsolved problem.

line 74: There are several conditions which could result in unsolved problems such as those illustrated in  Figures 4a and 4b which are the most common.

line 75: In order to get better localization results, the UWB ranging errors should be addressed.

line 78: In the UWB distance estimation, the obstacle between the UWB transmitter and receiver has to avoid the line-of-sight propagation of the UWB signal.

line 80: The estimated distance in NLOS conditions …

line 84: … UWB ranging distance data. Both the state and observation equations are as follows:

line 85: … are the observation and process noise, respectively and …

line ?: The mean square error of the next step prediction is

line 86: ¿?

line ?: The gyroscope is one of the key devices in INS localization and the main error is due to the gyroscope drifting. In short time, the white and quantizing noises cause an important effect on the angular error. The modeling of the gyroscope mainly includes: constant and random errors and the model of the gyroscope is:

line ?: In the design of the integrated localization method, the biased error Di (i = x, y, z) is considered in the error model of the accelerometer:

line 89: The error of the INS localization includes the error of the device, which consist of the scale and the installation errors.

line 91: The principle of the EKF is to linearize the system, and transfer the system model to state equation, measure the equation error, and predict the state error with the standard Kalman filter.

line 94: where h(x) is the nonlinear function of the state vector.

line 95: After linearization, standard Kalman filtering is used to calculate both the error state and the observation models.

line ?: At the same time, the system will record

line 99: With the UWB ranging results RN and the coordinates of the UWB anchor nodes Pn, the accurate localization result could be achieved with EKF integrating the INS data.

line 101: … can be represented …

line 78: … can be obtained from…

line ?: In the warehouse scenario, once the position coordinate [xk, yk], the velocity npn and the angle between the AGV and +y direction of the coordinate are known, the left and right side positions can be calculated from the size of AGV as follows,

where w is the width of the AGV.

line 108: … delay of the AGV and assuming that …

line 111: If the area covering the four points [lxk , lyk ], [rxk , ryk ], [lxk+1 , lyk+1 ], [rxk+1 , ryk+1] does not contain obstacles in the electronic map, no collision happens.

4. Experiment results

line 118: It is a small and multi-port AGV high reliability robot system.

line 121: … anchor nodes and the triangle represents the sample test locations.

line 123: The error distribution is shown in Fig. 8b. It is obvious that the test points closer to the center of the three UWB anchor nodes achieve smaller positioning error.

line 125: According to the experimental results, when the UWB anchor nodes are far from each other, the positioning accuracy is poor.

line 127: The results are meaningful for the deployment of the UWB anchor nodes and users can choose …

line 136: Three positioning methods including UWB, INS and INS-UWB are tested and compared. The results are shown in Fig. 10.

line 140: The error of the INS positioning is smaller in the first period and larger in the middle and last periods. The INS-UWB positioning method has the advantages of both positioning methods.

line 142: The overall error is smaller than both single positioning methods and it is relatively stable, less than 15 cm, meeting the requirement for AGV positioning.

line 144: … instructed to cross the warehouse with shelves in order to reach a destination.

line 145: The collision avoidance effect is shown in Figure 11 where the black blocks are the obstacles and the red line is the final route of the AGV.

line 146: According to the results, the route in the blue circles represents that the AGV has correctly detected the obstacles and it has correctly adjusted its direction to avoid the collision.

line 146: The experimental results show that the INS-UWB based AGV

collision avoidance system can be used in environments with fixxed obstacles.

5. Conclusion

line 150: 5. Conclusions

line 151: Compared to traditional inertial- and laser-based guiding systems, the proposed method is more flexible and cheaper with good accuracy and stability.

line 153: In order to realize an accurate positioning, an electronic map of the warehouse is established where the UWB anchor nodes are deployed.

line 155: The location coordinate of the AGV is obtained by three nearby UWB anchor nodes in the map.

line 157: In order to avoid collisions, the current location of …

line 160: Experimental results show that the method proposed in this paper achieves accurate positioning estimation and the AGV effectively detects obstacles and avoids possible collisions thus, ensuring the safety of the AGV.

Author Response

Dear Reviewers:

Thank you for your comments concerning our manuscript entitled An INS-UWB based collision avoidance system for AGV. Those comments are all valuable and very helpful for revising and improving our paper, as well as the important guiding significance to our researches. We have studied comments carefully and have made correction which we hope meet with approval

Round 3

Reviewer 1 Report

The work presented in this paper is interesting, the overall merit is clear and the results are promising.

Author Response

Dear Academic Editors and Reviewer:

Thank you for your letter and for the comments concerning our manuscript entitled An INS-UWB based collision avoidance system for AGV. Those comments are all valuable and very helpful for revising and improving our paper, as well as the important guiding significance to our researches. We have studied comments carefully and have made correction which we hope meet with approval.